# A Prospective, Randomized, Comparative Study of Topical Minocycline Gel 4% with Topical Clindamycin Phosphate Gel 1% in Indian Patients with Acne Vulgaris

**DOI:** 10.3390/antibiotics12091455

**Published:** 2023-09-19

**Authors:** Bela Shah, Deval Mistry, Nelry Gonsalves, Presha Vasani, Dhiraj Dhoot, Hanmant Barkate

**Affiliations:** 1Department of Dermatology, B J Medical College, Civil Hospital, Ahmedabad 380016, Gujarat, India; shah.drbela@gmail.com (B.S.); gonsalvesnelry@gmail.com (N.G.); preshavasani@gmail.com (P.V.); 2Consultant Dermatologist, Mistry Skin Clinic, Ahmedabad 301302, Gujarat, India; deval.neogeo@gmail.com; 3Department of Global Medical Affairs, Glenmark Pharmaceuticals Ltd., Mumbai 400099, Maharashtra, India; hanmant.barkate@glenmarkpharma.com

**Keywords:** acne vulgaris, inflammatory, non-inflammatory lesions, investigator’s global assessment score, minocycline, clindamycin, efficacy, safety

## Abstract

Acne vulgaris is characterized by inflammatory and non-inflammatory skin lesions with a high prevalence among adolescents in India. Not enough studies are reported on the use of topical antibiotics for the management of acne in the Indian population. The proposed study aims to compare the efficacy and safety of topical minocycline gel 4% with topical clindamycin gel 1% in the Indian population. A randomized, open-label, double-arm study was planned at two centers in India. One hundred patients were enrolled and randomized equally to two treatment arms. The drugs were applied once daily, preferably at the same time each day. The number of inflammatory and non-inflammatory lesions, as well as the investigator’s global assessment (IGA), were obtained at the baseline and on weeks 3, 6, 9, and 12. The change in these parameters from baseline to week 12 was compared between the two treatment arms. A tolerability assessment was also performed on selected parameters. The age of patients ranged between 14 and 31 years, with female preponderance in each arm. On week 12, the percent change in inflammatory and non-inflammatory lesions in the minocycline 4% arm was significantly higher than in the clindamycin 1% arm (*p* < 0.0001). The IGA treatment success was significantly higher in the minocycline arm compared to the clindamycin arm on weeks 9 and 12, with *p*-values of 0.001 and 0.015, respectively. Tolerability assessment revealed significantly improved parameter performance in the minocycline arm compared to the clindamycin arm. On subgroup analysis, in adolescents, minocycline was found to be more efficacious than clindamycin. The comparative assessment resulted in a significantly improved performance of minocycline gel 4% compared to clindamycin gel 1% in the Indian population, thus making it a preferred choice for the treatment of moderate-to-severe acne in India.

## 1. Introduction

Acne vulgaris is a highly prevalent skin disorder characterized by the presence of both inflammatory and non-inflammatory lesions. It affects a substantial portion of the population, and about 85% of the population experiences acne at some stage of their lives. The prevalence of acne varies from 35% to almost 100% among adolescents in different countries [1,2]. In India, there are few small, single-centric studies studying clinico-epidemiological aspects of acne and factors aggravating or precipitating acne [3]. Acne is a chronic condition triggered by hormonal changes during adolescence and gets further aggravated due to genetic factors. The sebaceous glands play a major role in the disease manifestation, as they have all the mechanisms for hormone and cytokine production [4]. Moreover, mental and emotional stress alters the tissue environment in the pilo-sebaceous follicle, thereby initiating central and local expression of neuropeptides, causing the disease condition [5].

Orally administered antibiotics play a key role in the management of acne vulgaris, and although antibiotics like oxytetracycline, erythromycin, and minocycline are well tolerated, prolonged usage of antibiotics for the treatment is not recommended due to resistance [6]. Hence, over the years, several topical antibiotics have been introduced for the treatment of acne as an alternative to systemic antibiotic therapy. Clindamycin and erythromycin are the most prescribed topical antibiotics, but an increase in resistance against them has curtailed their use [7,8]. While oral antibiotics like minocycline are effective for the management of acne, they should be used cautiously due to systemic adverse events. Hence, there is a need for existing systemic antibiotics like minocycline, which is effective against acne, to be used in a topical formulation.

Minocycline is a semisynthetic second-generation tetracycline. It has anti-inflammatory and bacteriostatic properties and has been shown to be effective in treating moderate-to-severe acne [9]. In 2019, the US FDA approved minocycline 4% foam for the treatment of acne of a moderate-to-severe intensity [10]. The use of this foam formulation aims to circumvent the systemic side effects associated with the oral administration of minocycline while retaining its anti-inflammatory and bacteriostatic activity against the target anaerobic microbes [9,11]. In a long-term study, 4% minocycline foam appeared to be safe, effective, and well tolerated for up to 52 weeks for the treatment of moderate-to-severe acne [11]. Subsequently, in 2022, minocycline gel was approved in India by the Drugs Controller General of India (DCGI) [12]. Owing to the thermo-stability concern of the foam, a gel formulation was approved.

Minocycline hydrochloride topical gel 4%, being a new topical formulation, has not yet been studied on Indian patients, especially in comparison to established antibiotics like clindamycin. Therefore, the objective of this study was to evaluate the efficacy and safety of topical minocycline gel 4% against clindamycin phosphate gel 1% in the management of acne vulgaris among the Indian population.

## 2. Results

### 2.1. Patient Disposition

A total of 100 patients were enrolled in the study from two centers and were randomized equally into two arms. During the study duration, four patients from the topical minocycline 4% arm and five patients from the topical clindamycin 1% arm were lost to follow-up. The final analysis set comprised 46 patients from the topical minocycline 4% arm and 45 patients from the topical clindamycin 1% arm (Figure 1).

### 2.2. Demographic and Disease Characteristics at Baseline

The baseline characteristics of patients from the two arms were compared, as shown in Table 1. The mean age of patients in the topical minocycline 4% arm was 19.76 ± 4.38 years, while that of the topical clindamycin 1% arm was 20.58 ± 4.59 years. The difference in means between the two arms was statistically non-significant. Similarly, the difference in the distribution of patients by gender and grade of acne were statistically non-significant. The mean duration of acne, the number of inflammatory and non-inflammatory lesions, and the investigator’s global score also differed non-significantly between the two arms, suggesting a homogeneous distribution.

### 2.3. Efficacy Analysis

Out of 50 patients recruited in each treatment arm, 46 (92%) from the topical minocycline 4% and 45 (90%) from the topical clindamycin 1% arms completed the study. The number of inflammatory and non-inflammatory lesions were compared between the two arms during each visit, as shown in Table 2. At the baseline, the median number of inflammatory lesions in the minocycline 4% arm was higher than that of the clindamycin arm; however, the difference was statistically non-significant. On weeks 9 and 12, the medians were significantly lower in the minocycline 4% arm compared to the clindamycin 1% arm, with *p*-values of 0.047 and 0.038, respectively. The change in the number of inflammatory lesions from baseline to week 12 was statistically significant in both treatment arms (*p* < 0.0001). The percent reduction in the number of inflammatory lesions from baseline to different visits was obtained, as illustrated in Figure 2. The percent change was significantly higher in the minocycline 4% arm compared to the clindamycin 1% arm for all the visits. On week 12, the percent change in the minocycline 4% arm (−88.5%) was significantly higher than that of the clindamycin arm (−65.81%) with a *p* < 0.0001.

Along similar lines, the numbers of non-inflammatory lesions were compared between the two arms during different visits. From week 6 onwards, the median number was significantly smaller in the minocycline 4% arm compared to the clindamycin 1% arm (*p* < 0.05). The change in the median number of non-inflammatory lesions from baseline to week 12 was statistically significant in both treatment arms (*p* < 0.0001). Further, the percent reduction in the number of non-inflammatory lesions from baseline to different visits was significantly higher in the minocycline 4% arm compared to the clindamycin 1% arm (Figure 2). On week 12, the percent change in the minocycline 4% arm (−87.8%) was significantly higher than that of the clindamycin 1% arm (−63.59%) with a *p* < 0.0001.

The comparison of the investigator’s global assessment scores was performed between the arms during different visits (Table 3). Only on week 9 was the median score significantly lower in the minocycline 4% arm compared to the clindamycin 1% arm (*p* = 0.001), while for other visits, the difference was non-significant. The reduction in the score from baseline to week 12 was statistically significant in both the treatment arms (*p* < 0.0001). The investigator’s global assessment treatment success was significantly higher in the minocycline 4% arm compared to the clindamycin 1% arm on weeks 9 and 12 with *p*-values of 0.001 and 0.015, respectively (Table 3). There were 39 (84.8%) patients in the minocycline 4% arm and 28 (62.3%) in the clindamycin 1% arm who either cleared or almost cleared acne on week 12.

### 2.4. Efficacy among Young Adults and Adolescents

The comparison of the percent reduction in inflammatory and non-inflammatory lesions from baseline to week 12 was performed between young adult (aged between 19 and 34 years) and adolescent (aged between 10 and 19 years) subjects in each treatment arm [13] (Table 4). The difference in percent reduction in both the type of lesions between young adult and adolescent groups was statistically non-significant. However, the between-arms comparison in adolescents revealed a significantly higher reduction in inflammatory lesions in the topical minocycline 4% arm compared to the topical clindamycin 1% arm (*p* = 0.004). Similarly, the percent reduction in non-inflammatory lesions was significantly higher in the topical minocycline 4% arm compared to the topical clindamycin 1% arm (*p* < 0.0001) in adults as well as adolescents. Further, the change in the IGA score from baseline to week 12 showed a statistically non-significant difference between young adults and adolescents in both the treatment arms. However, in young adults, the change in IGA score was significantly higher in the topical minocycline 4% arm compared to the topical clindamycin 1% arm (*p* = 0.001). Among adolescents, the change in IGA score was also significantly higher in the topical minocycline 4% arm compared to the topical clindamycin 1% arm (*p* = 0.037).

### 2.5. Safety and Tolerability Assessment

Dryness of skin showed significantly different results between the two treatment arms during all the visits, with the number of mild cases in the minocycline 4% arm significantly lower than that of the clindamycin 1% arm (Table 5). Skin peeling was present in the milder form in both the treatment arms. However, in the minocycline 4% arm, on week 6, the proportion was significantly lower compared to the clindamycin 1% arm (*p* = 0.014); upon the continuation of clindamycin, on week 12, the result was non-significant. Itching was significantly lower in patients from the minocycline 4% arm compared to the clindamycin 1% arm on weeks 9 and 12 (*p* < 0.05). Other parameters, such as erythema and hyperpigmentation, differed non-significantly between the two arms. The change in the lesions before and after treatment in the minocycline 4% arm is shown in Figure 3.

## 3. Discussion

Minocycline, an oral antibiotic of the second-generation tetracycline class with the lowest rate of resistance and anti-inflammatory and bacteriostatic activities, is a preferred first-line therapy for the treatment of moderate-to-severe acne vulgaris [14,15,16,17]. Minocycline foam was formulated as micronized minocycline hydrochloride crystals suspended in an oleaginous foam base [18]. Another commonly prescribed topical antibiotic for acne treatment is clindamycin, which has proven its anti-inflammatory properties and effectiveness in reducing lesions [19]. A topical clindamycin 1% was formulated by nano-emulsion technology that facilitates a larger surface area than other gels, thereby providing better penetration into the pilo-sebaceous glands [20]. In the early 90s, a double-blind comparison of oral minocycline and topical clindamycin demonstrated equivalence, suggesting the latter was an effective substitute for oral minocycline [6]. However, prolonged use of topical clindamycin as well as erythromycin has resulted in drug resistance and even cross-resistance to different antibiotics [7,21,22]. In comparison, minocycline demonstrated lower rates of antibiotic resistance and low cross-reactivity [7]. In 2019, a topical formulation of minocycline (FMX101 4% foam) received approval for the treatment of moderate-to-severe acne vulgaris in patients above 9 years old [9]. This foam formulation aimed to circumvent the systemic side effects associated with the oral administration of minocycline while retaining its anti-inflammatory and bacteriostatic activity against the target anaerobic microbe *C. acnes* [11]. Topical minocycline 4%, being a new formulation, has not been studied on Indian patients, especially in comparison to established topical antibiotics like clindamycin. Hence, a study was designed to evaluate the efficacy, safety, and tolerability of topical minocycline 4% in comparison to topical clindamycin 1% in the Indian population. It is worth noting that gel formulation is favored over foam formulation in India due to concerns regarding its thermo-stability.

Starting from week 6, the topical minocycline exhibited significantly better results in non-inflammatory lesions compared to the topical clindamycin. Additionally, from week 9 onwards, the topical minocycline exhibited a significantly lower number of inflammatory lesions compared to the topical clindamycin. On week 12, the mean percent reduction in inflammatory and non-inflammatory lesions from the baseline was significantly higher in topical minocycline compared to topical clindamycin. Although there are no comparative studies using these two treatments, in a study using topical minocycline 4% against the vehicle in the Israeli population, the authors reported a significantly higher reduction in inflammatory lesions (−71.7% vs. −50.6; *p* = 0.0001) and non-inflammatory lesions (−72.7% vs. −56.5%; *p* = 0.019) in the test arm from the baseline to week 12 [23]. In another similarly designed study, the authors reported a significantly higher reduction in inflammatory lesions in the topical minocycline 4% arm against the foam vehicle (−56% vs. −43%; *p* < 0.0001) and a higher reduction in non-inflammatory lesions (−39% vs. −33%; *p* = 0.0036) [15]. Similarly, Gold LS et al. also reported a statistically significant reduction in inflammatory and non-inflammatory lesions from the baseline to week 12 compared to a foam vehicle [9].

Throughout the initial 6 weeks of the study, the Investigator’s Global Assessment (IGA) scores were similar in both arms. However, on week 9, the median score in the topical minocycline 4% arm was significantly lower than that in the topical clindamycin 1% arm (*p* = 0.001). Shamer A. et al. observed a significantly higher IGA treatment success rate in topical minocycline 4% compared to the foam vehicle on week 9 and onward [23]. In studies by Gold LS et al. and Raoof TJ. et al., the authors observed statistically superior success rates with topical minocycline 4% compared to the foam vehicle [9,16].

In the sub-group analysis, minocycline was statistically better than clindamycin in reducing inflammatory and non-inflammatory lesions as well as in achieving IGA success in adolescents. This suggests that minocycline may be preferred over clindamycin when choosing antibiotics in the treatment of adolescent acne.

Moreover, in our study, by the end of week 12, topical minocycline 4% exhibited better tolerability compared to topical clindamycin 1%. Although tolerability indicators were in the mild form in both the treatment arms, the proportions of mild cases were less in the topical minocycline 4% arm. Dryness was a notable concern in the topical clindamycin 1% arm, which could be attributed to the formulation. Overall, both the treatment arms demonstrated improvement in primary and secondary endpoints. Nevertheless, topical minocycline 4% statistically outperformed topical clindamycin 1%. Earlier studies established the superiority of topical minocycline 4% over the foam vehicle, and now, to supplement them, our study showed that it is more efficacious than topical clindamycin 1% as well.

Antibiotic resistance is often a concern while prescribing drugs. There is growing evidence of topical clindamycin and even erythromycin resistance in the past few decades [24]. On the contrary, the tetracycline class of antibiotics and minocycline has demonstrated the lowest rates of antibiotic resistance [10]. Topical minocycline 4% has been shown to have significantly lower systemic absorption, thereby reducing the risk of systemic bacterial resistance [10].

The observations made in the study are subject to the limitation of sample size. Conducting a study with a larger sample size and a robust study design like a double-dummy technique or split-face comparisons could provide further insights into the benefits of topical minocycline formulation over other topical alternatives. An extended trial duration beyond 12 weeks could also shed more light on the therapeutic role of the formulation.

## 4. Materials and Methods

### 4.1. Study Design

A prospective, randomized, open-label, double-arm clinical study was conducted to assess the efficacy, safety, and tolerability of topical minocycline gel 4% against clindamycin phosphate gel 1% in Indian patients with acne vulgaris. After obtaining written informed consent, patients were screened, and eligible patients with acne vulgaris were enrolled in the study. Demographic information and medical history of the patients were recorded. The patients were scheduled to visit every 3 weeks up to week 12. For efficacy evaluations, acne lesion count (inflammatory and non-inflammatory lesions) and investigator’s global assessment (IGA) were performed at baseline and on weeks 3, 6, 9, and 12, with the aim of determining changes in the inflammatory and non-inflammatory lesion count from baseline. The IGA scale, which involves a grading system ranging from 0 (clear), 1 (almost clear), 2 (Mild severity), 3 (Moderate severity), to 4 (Severe), was used for the assessment [25]. Treatment success was defined as an improvement of at least 2 grades in the IGA score from the baseline, along with an IGA score of either 0 or 1. Safety was assessed using local skin tolerability on 4-point Likert scale (0: None; 1: Mild; 2: Moderate; 3: Severe) for patient-reported symptoms and those recorded by the treating physician [26].

### 4.2. Participant’s Eligibility Criteria

The study included patients aged 9 years and above, of either gender, diagnosed with acne vulgaris, who agreed to abstain from using any other acne medications or medicated cleansers, refrain from excessive sun exposure during the study period, and were willing to participate. Patients with severe systemic diseases, facial sunburn, pregnant and lactating females, allergic to tetracycline-class of drugs, clinically significant hepatic impairment, usage of systemic antibiotics, and those receiving investigational therapy, including COVID-19 vaccine within 28 days of screening, were excluded from the study. A total of 100 patients diagnosed with acne vulgaris and meeting the eligibility criteria were randomized in 1:1 ratio to the treatment arms, using block randomization with a block size of two. Patients were instructed to adhere to the predefined visit schedule as outlined in the study protocol.

### 4.3. Drug Administration

Both the drugs were applied once daily, preferably at the same time every day in the evening. Patients were instructed to use a small amount of topical gel and apply it to the areas of the face affected by acne. This process was repeated as needed until all acne-affected parts of the face were treated. For acne present on other parts of the body (neck, shoulders, arms, back, or chest), an additional amount of topical gel was applied by the patient. The patients were asked not to bathe, shower, or swim for at least 1 h after application of the product. Patient compliance was assessed by collecting and inspecting the empty tubes of topical gel during each visit.

### 4.4. Efficacy and Safety Endpoints

The primary endpoints of the study were to compare the change in the number of inflammatory and non-inflammatory lesions from baseline to weeks 3, 6, 9, and 12 between the two treatment arms. Additionally, another primary endpoint was to compare the investigator’s global assessment (IGA) score and the proportion of IGA success between the two arms during different visits. The secondary endpoint of the study was to compare the local skin tolerability of patients in the two treatment arms. The safety endpoint comprised of treatment-related adverse/serious adverse events leading to discontinuation of the treatment.

### 4.5. Statistical Methods

The continuous parameters, the number of inflammatory and non-inflammatory lesions, and the scores were expressed in terms of mean, median, standard deviation, minimum, and maximum, while the categorical parameters were expressed in terms of frequencies and percentages. The parameters following normal distribution according to Shapiro–Wilk’s test were compared between arms using *t*-test for independent samples. The gender and grade of acne were compared using Pearson’s chi-square test. The number of inflammatory and non-inflammatory lesions were compared between two arms at different time points using Mann–Whitney U test, while the changes in the number of lesions within arm were compared using Friedman analysis of variance. The change in the number of lesions from baseline to different time points was compared between two arms using Mann–Whitney U test. The analysis of investigator’s global assessment score was also carried out along similar lines. The comparison of investigator’s global assessment treatment success and tolerability assessment was performed using Pearson’s chi-square test. The acne quality of life at baseline and week 12 was compared between two arms using Mann–Whitney U test, while within-arm comparison between two time points was performed using Wilcoxon signed rank test. All the analyses were performed using SPSS ver. 26.0 (IBM Corp., Armonk, NY, USA), and the statistical significance was tested at 5% level.

## 5. Conclusions

To the best of our knowledge, in the Indian population, this is the first comparative investigation examining the efficacy, safety, and tolerability of a novel topical minocycline gel 4% formulation in comparison to topical clindamycin gel 1% for the treatment of moderate-to-severe acne. The results of primary endpoints indicate a statistically significant improvement with the use of topical minocycline 4% when compared to topical clindamycin 1%. Additionally, in adolescents, minocycline gel was found to be more efficacious than clindamycin in clearing acne. Moreover, topical minocycline 4% appeared to be safe and well tolerated relative to topical clindamycin 1%. Topical minocycline 4% can be considered a preferable choice for the treatment of individuals with moderate-to-severe acne in India.

## Figures and Tables

**Figure 1 antibiotics-12-01455-f001:**
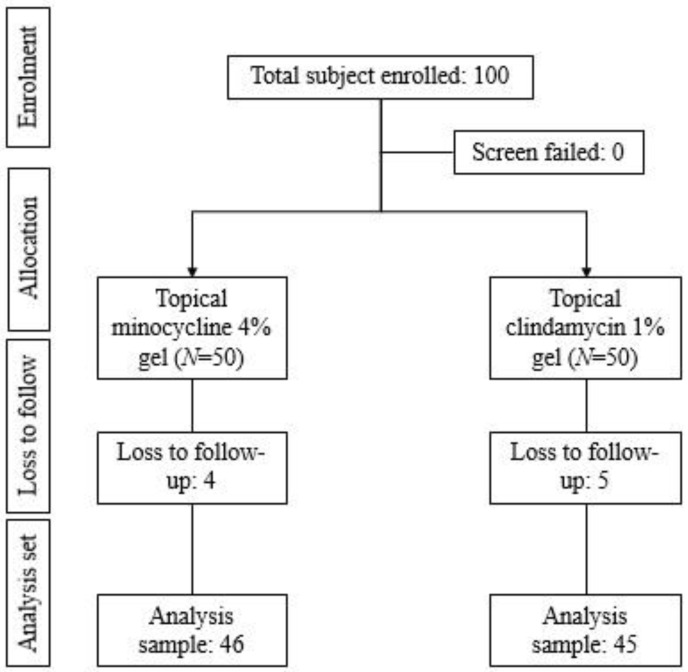
Patient flow in study.

**Figure 2 antibiotics-12-01455-f002:**
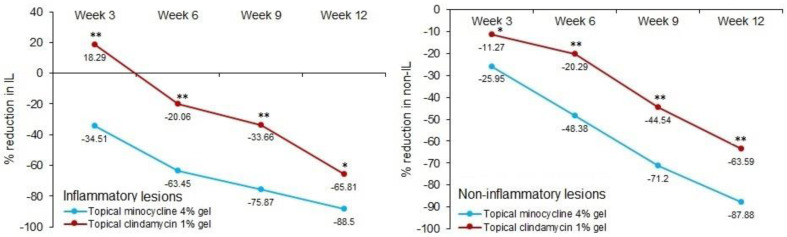
Percent reduction in inflammatory and non-inflammatory lesions according to time in two treatment groups (* *p* < 0.05; ** *p* < 0.0001).

**Figure 3 antibiotics-12-01455-f003:**
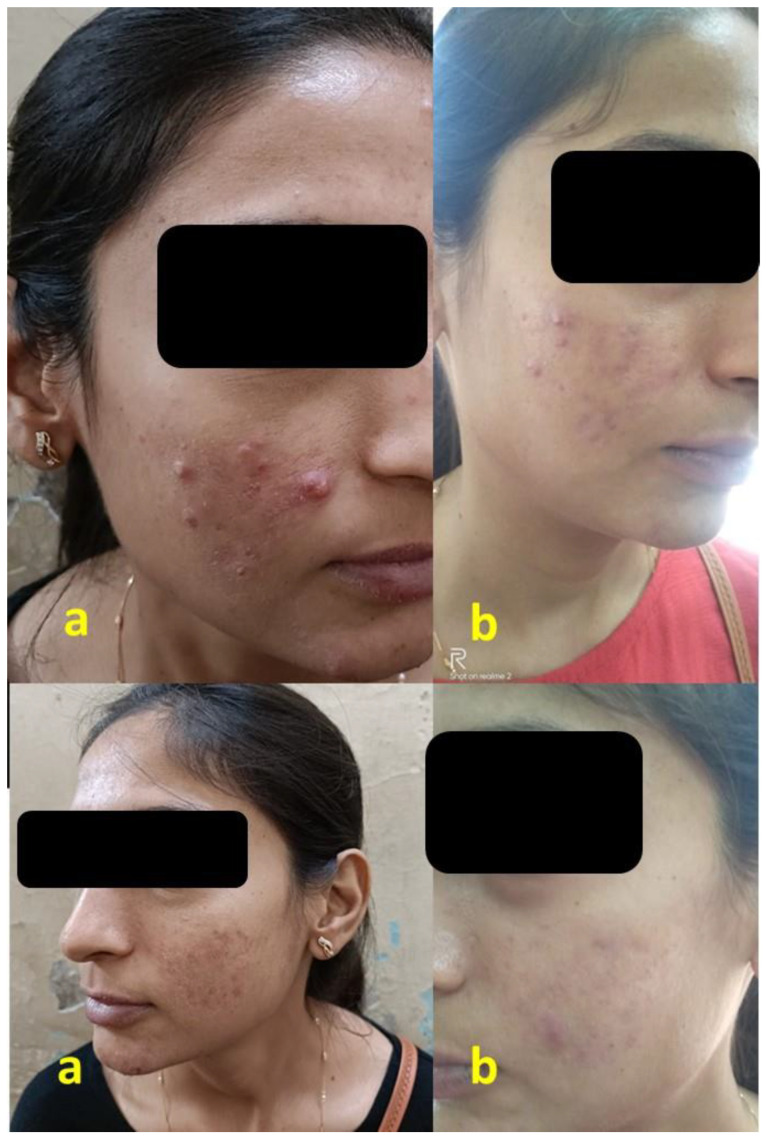
Clinical efficacy of minocycline gel 4%; (**a**) before treatment and (**b**) after treatment.

**Table 1 antibiotics-12-01455-t001:** Descriptive statistics for demographic and disease characteristics of patients at baseline in two treatment groups.

Characteristics		Treatment		*p*-Value
		Topical Minocycline	Topical Clindamycin	
		4% Gel (*N* = 50)	1% Gel (*N* = 50)	
Age (years) ^1^		19.76 ± 4.38; 18.0; (14, 31)	20.58 ± 4.59; 20.0; {14, 32)	0.363 †
Gender ^2^	*Male*	18 (36.0)	16 (32)	0.673 *
	*Female*	32 (64.0)	34 (68)	
Grade of acne ^2^	*1*	10 (20.0)	18 (36.0)	0.069 *
	*2*	31 (62.0)	29 (58.0)	
	*3*	9 (18.0)	3 (6.0)	
Duration of acne (months) ^1^		12.24 ± 13.63; 7.00; (1, 60)	8.08 ± 9.91; 4.00; (1, 48)	0.084 †
Number of inflammatory lesions ^1^		6.22 ± 5.03; 5.00; (0, 20)	4.32 ± 3.89; 3.00; (0, 15)	0.066 ‡
Number of non-inflammatory lesions ^1^		14.06 ± 8.22; 12.00; (2, 42)	12.74 ± 6.21; 12.00; (0, 35)	0.618 ‡
Investigator’s global score ^1^		2.64 ± 0.9; 3.00; (1, 5)	2.44 ± 1.15; 2.00; (1, 9)	0.059 ‡

^1^ Data expressed as mean ± SD; median; (min, max); ^2^ expressed as *n* (%); † obtained using *t*-test for independent samples; * obtained using Pearson’s chi-square test; and ‡ obtained using Mann–Whitney U test.

**Table 2 antibiotics-12-01455-t002:** Comparison of inflammatory and non-inflammatory lesions between two groups (ITT sample).

Parameter	Visit	Treatment								*p*-Value ^2^
		Topical Minocycline				Topical Clindamycin				
		4% Gel (*N* = 50)				1% Gel (*N* = 50)				
		*n*	Mean	SD	Median	*n*	Mean	SD	Median	
Number of inflammatory lesions	Baseline	50	6.22	5.03	5	50	4.32	3.89	3	0.066
	Week 3	50	4.54	4.36	4	50	4.48	3.59	4	0.7
	Week 6	50	2.8	3.02	2	50	3.36	2.96	3	0.221
	Week 9	49	1.8	2.26	1	49	2.71	2.61	2	**0.047**
	Week 12	46	0.98	1.51	0	45	1.69	1.84	1	**0.038**
	*p*-value ^1^	**<0.0001**				**<0.0001**				
Number of non-inflammatory lesions	Baseline	50	14.06	8.22	12	50	12.74	6.21	12	0.618
	Week 3	50	10.42	6.27	10	50	10.8	4.9	10	0.274
	Week 6	50	7.52	5.1	7	50	9.26	4.21	8	**0.01**
	Week 9	49	4.39	3.7	4	49	6.69	3.7	6	**0.001**
	Week 12	46	2.07	2.45	1.5	45	4.49	2.84	4	**<0.0001**
	*p*-value ^1^	**<0.0001**				**<0.0001**				

^1^ Obtained using Friedman ANOVA and ^2^ obtained using Mann–Whitney U test. Bold *p*-value indicates statistical significance.

**Table 3 antibiotics-12-01455-t003:** Comparison of investigator’s global assessment parameters between two groups (ITT).

Parameter	Visit	Treatment								*p*-Value ^2^
		Topical Minocycline				Topical Clindamycin				
		4% Gel (*N* = 50)				1% Gel (*N* = 50)				
		*n*	Mean	SD	Median	*n*	Mean	SD	Median	
Investigator’s global assessment score	Baseline	50	2.64	0.9	3	50	2.44	1.15	2	0.059
	Week 3	50	2.20	0.86	2	50	2.26	0.56	2	0.808
	Week 6	50	1.76	0.85	2	50	2.02	0.59	2	0.080
	Week 9	49	1.22	0.65	1	49	1.71	0.65	2	**0.001**
	Week 12	46	0.74	0.71	1	45	1.02	0.87	1	0.113
	*p*-value ^1^	**<0.0001**				**<0.0001**				
Investigator’s Global Assessment—Treatment success at (YES)		*n* (%)				*n* (%)				*p*-value ^3^
	Week 3	0 (0)				0 (0)				-
	Week 6	4 (8.0)				0 (0)				0.126
	Week 9	15 (30.6)				2 (4.1)				**0.001**
	Week 12	34 (73.9)				21 (46.7)				**0.015**

^1^ Obtained using Friedman ANOVA; ^2^ obtained using Mann–Whitney U test; and ^3^ obtained using chi-square test. Bold *p*-value indicates statistical significance.

**Table 4 antibiotics-12-01455-t004:** Comparison of percent reduction in inflammatory and non-inflammatory lesions as well as IGA scores between adolescent and adult subjects within each group and for each subject category between groups.

Change from Baseline to 12 Months	Treatment														*p*-Value *^,1^ (Young Adult)	*p*-Value *^,2^ (Adolescent)
	Topical Minocycline 4% Gel							Topical Clindamycin 1% Gel								
	Age > 19 (Young Adult)			Age ≤ 19 (Adolescent)			*p*-Value *	Age > 19 (Young Adult)			Age ≤ 19 (Adolescent)			*p*-Value *		
	*n*	Mean	SD	*n*	Mean	SD		*n*	Mean	SD	*n*	Mean	SD			
% reduction in inflammatory lesions	20	88.24	13.89	26	88.71	14.47	0.988	24	66.90	36.41	21	64.60	30.54	0.645	0.086	**0.004**
% reduction in non-inflammatory lesions	20	87.30	14.59	26	88.32	10.35	0.899	24	65.00	21.16	21	61.91	21.10	0.697	**0.001**	**<0.0001**
IGA score	20	1.90	0.64	26	1.92	0.84	0.797	24	1.17	0.92	21	1.71	1.59	0.383	**0.001**	**0.037**

* Obtained for comparison between young adult and adolescent subjects within each treatment group; *^,1^ obtained for comparison of young adults between two groups; and *^,2^ obtained for comparison of adolescents between two groups. Bold values indicate statistically significant difference. *p*-value obtained using Mann–Whitney U test.

**Table 5 antibiotics-12-01455-t005:** Comparison of tolerability assessment parameters between two arms (ITT sample).

Tolerability Assessment Parameter	Visit	Topical Minocycline 4% Gel (*N* = 50)				Topical Clindamycin 1% Gel (*N* = 50)				*p*-Value
		None	Mild	Moderate	Intense	None	Mild	Moderate	Intense	
Erythema	Week 3	40	9	1	0	32	17	1	0	0.187
	Week 6	47	3	0	0	43	7	0	0	0.317
	Week 9	47	2	0	0	40	9	0	0	0.055
	Week 12	46	0	0	0	40	5	0	0	0.062
Dryness	Week 3	44	6	0	0	30	19	1	0	**0.005**
	Week 6	47	3	0	0	31	19	0	0	**<0.0001**
	Week 9	45	4	0	0	35	14	0	0	**0.019**
	Week 12	40	6	0	0	22	23	0	0	**<0.0001**
Hyperpigmentation	Week 3	50	0	0	0	50	0	0	0	-
	Week 6	49	1	0	0	50	0	0	0	0.999
	Week 9	45	4	0	0	49	0	0	0	0.126
	Week 12	44	2	0	0	45	0	0	0	0.484
Skin peeling	Week 3	44	6	0	0	38	12	0	0	0.193
	Week 6	45	5	0	0	34	16	0	0	**0.014**
	Week 9	43	5	1	0	38	11	0	0	0.169
	Week 12	45	1	0	0	38	7	0	0	0.060
Itching	Week 3	36	13	1	0	33	16	1	0	0.802
	Week 6	42	8	0	0	41	9	0	0	0.999
	Week 9	49	0	0	0	36	13	0	0	**<0.0001**
	Week 12	44	2	0	0	30	15	0	0	**0.001**

Obtained using Pearson’s chi-square test. Bold *p*-value indicates statistical significance.

## Data Availability

The datasets are available only on request due to privacy/ethical restrictions and can be requested from ddhoot@gmail.com.

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
