# Peer review of "A Prospective, Randomized, Comparative Study of Topical Minocycline Gel 4% with Topical Clindamycin Phosphate Gel 1% in Indian Patients with Acne Vulgaris"

_antibiotics, 2023, doi:10.3390/antibiotics12091455_

Round 1

Reviewer 1 Report

The paper is really a good paper that present a good study. The statistical data are presented in a good way. Two suggestions to take in consideration:

1- the word "group" to be used instead of the word "arm".

2- please indicate the use of antibiotic was made to the acne on moderate and severe stage referring the classification of acne vulgaris or you have use it either to the mild condition of acne?

Author Response

Thankd for reviewing the manuscript...All the suggestion ammended...

Reviewer 2 Report

the article I do not think is of high scientific content. 

The use of topical and oral antibiotics for acne are now well known in the literature, this new formulation of minocycline is not new ; plus why not start from the differences for oral ones? 

There are few parameters for evaluating pre and post

The future will involve combination of various molecules in one product 

Materials and methods are after discussion this confuses the reader

There is a lack of a materials and methods section 

I do not think it is a candidate for publication

Moderate editing of English language required

Moderate editing of English language required

Author Response

Thanks for reviewing the manuscript. All the queries/suggestions answered...

Reviewer 3 Report

Abstract

The abstract is succint and well structured. It emphasises the main idea of the study, while also stirring an interest in its target reading audience .  The text present a clear description of the materials and methods that were involvolved in this study and its results, thus conveying the study’s utility in future medical practice. Furthermore, there is a decidely academic use of english language and the authors present us with a fluent text. 

Introduction 

The introduction is coherent,  well documented and comprehensive, while explaining the rationale that led to this study.

Results

The result are divided into 5 parts. The first and second subchapter that regard ”patient distribution”, respectively „Demographic and disease characteristics at baseline imply a homogeneous distribution of patients into the two groups and justify why some subjects were excluded from the final analysis. Part 3 and 4 focus on the „Efficacy analysis, detailing all the stages of acne severity assesment and all the methods that were used, as well as the results . Part 5, the Safety and tolerability assessment” is similarly well structured and well formulated.  All the original tables and figures are are properly described and overall helpful.

Discussions

This part of the study is throughoutly documented, citing relevant publications from the last 10 years, but focusing mostly on fairly recent ones; it constructs a very clear idea about the debate concering the formula and efficacy of minocycline  4%. The discussions also highlight the need for further research on the subject, in order to be better acquinted with the properties of applying minocylcine 4% gel on acne lesions. 

Matherial and Methods

This part is also subdivided into 5 topics, all of them being coherent and extensively described. 

Conclusions 

The conclusions are eloquent, highlighting the key results of the study.

Overview 

In conclusion, this  article consists of valuable research which might improve current clinical practice and might motivate further research in the field, and extend the aplicability of topical minocycline.  Although the article is carefully edited and uses proper language, I would recommend minor changes regarding its structure before publishing:

Position „matherial and methods” chapter  before the results, so the article is easier to follow

Author Response

Thanks for reviewing the manuscript. 

Round 2

Reviewer 2 Report

In my opinion is not suitable for publication